# From Single-Round to Sequential: Building Stateful Interactive Medical Image Segmentation with SegVol and GRU Corrector

Chuanyi Huang[1][0009−0009−3223−0082], Jinlong Huang[1][0009−0001−6102−6576], and Lisheng Wang[1*][0000−0003−3234−7511]

School of Automation and Intelligent Sensing, Shanghai Jiao Tong University, Shanghai 200240, China
{lswang}@sjtu.edu.cn

**Abstract.** Medical image segmentation has advanced considerably with foundational models like the Segment Anything Model (SAM) and its medical variants, yet real-world clinical deployment remains constrained by heterogeneous imaging protocols, limited data generalization, and the inefficiency of manual interaction. While recent SAM-based frameworks (e.g., SAM2, MedSAM2) introduce memory-aware mechanisms, they still rely on dense re-encoding and lack targeted correction strategies. We propose **"From Single-Round to Sequential: Building Stateful Interactive Segmentation with SegVol and GRU Corrector"**, a lightweight framework that reformulates interactive segmentation as a sequential refinement process guided by uncertainty and error heuristics. Specifically, we design: (1) a GRU-based temporal module to encode interaction history and enable stateful correction, and (2) an uncertainty-driven region adaptation scheme that selectively focuses refinement on ambiguous or mis-segmented areas, reducing redundant computation while improving correction efficiency. On validation data, our framework achieves a progressive Dice coefficient improvement from 0.661 (single-box prompt) to 0.671 after three refinement rounds, showing a 1.5% absolute gain with diminishing returns in later interactions. These results highlight that uncertainty-guided, memory-efficient refinement offers a promising direction for practical interactive medical segmentation.

**Keywords:** Interactive Medical Image Segmentation · Uncertainty-Driven Refinement · Sequential State Modeling · Lightweight GRU Correction

## 1 Introduction

### 1.1 Background

Medical image segmentation has undergone a paradigm shift with the emergence of foundation models such as the Segment Anything Model (SAM) [5] and its medical extensions [6,8]. By leveraging large-scale pretraining, these

models demonstrate remarkable generalization across diverse anatomical structures. However, their clinical translation remains hindered by substantial domain heterogeneity—multi-center imaging data vary widely in acquisition protocols, contrast levels, and patient populations [8]. Existing segmentation frameworks still struggle with small or irregular structures that violate shape priors and intensity uniformity [3]. Such limitations undermine clinical tasks that rely on accurate lesion boundaries, including quantitative diagnosis, treatment response assessment, and longitudinal monitoring [1]. These challenges underscore the need for interactive and adaptable segmentation systems that can efficiently integrate human feedback while remaining computationally feasible for routine use.

### 1.2    Related Work and Limitations

Interactive segmentation offers a promising way to bridge the gap between automated inference and clinical reliability. Foundation models such as SAM/SAM2 [5,9] and MedSAM/MedSAM2 [6,8] have demonstrated the potential of general-purpose visual prompts for medical imaging. Notably, recent versions already incorporate *memory-aware processing*, enabling limited contextual understanding across multiple prompts. Yet, their refinement process remains computationally expensive, as each correction still triggers dense re-encoding of the entire volume. Moreover, these models treat all regions uniformly during refinement, without explicitly prioritizing areas of high uncertainty or segmentation error. This results in redundant computation and limited gains with repeated interaction, particularly in volumetric 3D settings where localized corrections are more desirable.

Beyond SAM-based designs, frameworks such as VISTA3D [3] and nnInteractive [2] improve volumetric reasoning but largely preserve the stateless nature of interaction, handling each prompt independently. Consequently, they struggle to capture users' sequential intent—how each feedback relates to prior corrections—and fail to balance efficiency with accuracy [1]. These gaps motivate a new perspective: treating interactive segmentation not as isolated refinements, but as a sequential decision process guided by uncertainty and prior state information.

### 1.3    Contributions

To address these limitations, we propose **"From Single-Round to Sequential: Building Stateful Interactive Segmentation with SegVol and GRU Corrector"**, a lightweight framework that models interactive refinement as a temporally coherent process. Our main contributions are as follows.

First, we introduce a **GRU-based temporal corrector** that encodes the history of user interactions and segmentation states. Instead of reprocessing the full volume after each prompt, the model selectively updates predictions based on the accumulated context, effectively capturing user intent over time.

Second, we develop an **uncertainty-driven region adaptation** mechanism that heuristically identifies ambiguous or error-prone regions for targeted

correction. By focusing computation on these uncertain areas—typically near boundary inconsistencies or low-confidence voxels—the model reduces redundant processing while maintaining segmentation precision. This makes our approach particularly efficient for repeated interactions in complex cases such as irregular tumor margins or fine vascular networks.

Through these designs, our framework enables progressive, memory-efficient refinement in a stateful manner, achieving effective correction with minimal overhead. Unlike existing memory-heavy systems, our approach leverages uncertainty as an implicit guide for interaction, providing a practical and interpretable path toward adaptive clinical segmentation.

## 2   Method

Our framework, illustrated in Fig. 1, reformulates interactive medical image segmentation as a sequential refinement process. It integrates three main components: (1) **Box-Initialized Segmentation**, (2) **Uncertainty-Aware Interaction Sampling**, and (3) **GRU-Based Sequential Correction**. Together, these modules enable efficient, memory-aware refinement by focusing computation on uncertain and error-prone regions.

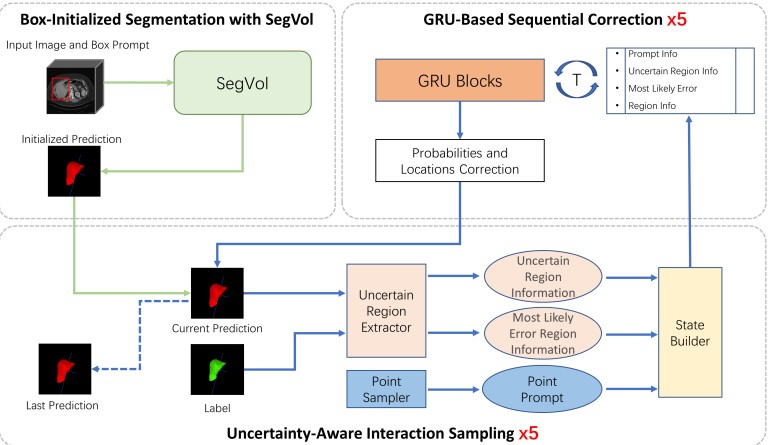

Fig. 1: Overview of the proposed framework. (a) SegVol produces an initial mask from a box prompt. (b) Uncertain and erroneous regions are sampled to build the sequential state tensor. (c) The GRU Corrector refines predictions based on accumulated interaction history.

## 2.1   Box-Initialized Segmentation with SegVol

Given a volumetric medical image $\mathbf{I} \in R^{D \times H \times W}$ and a user-specified bounding box $\mathbf{B} \in R^6$, the SegVol module generates an initial probability map $\mathbf{P}_0 \in [0,1]^{D \times H \times W}$ and a corresponding binary mask:

$$\mathbf{M}_0 = \mathrm{SegVol}(\mathbf{I}, \mathbf{B}), \quad \hat{y}_0 = I(\mathbf{M}_0 > 0.5), \tag{1}$$

where $I(\cdot)$ denotes the indicator function and $\hat{y}_0$ is the binarized segmentation output. This serves as the initial state for interactive refinement.

## 2.2   Uncertainty-Aware Interaction Sampling

To avoid reprocessing the full volume after every interaction, we design a lightweight sampling mechanism that identifies only the regions most in need of correction.

At iteration $t$, the current prediction produces a voxel-wise probability map $\mathbf{P}_t = \sigma(\mathbf{Z}_t)$, where $\mathbf{Z}_t$ denotes the raw logits from SegVol and $\sigma(\cdot)$ the sigmoid activation. We define the voxel-wise probability $p_t(i, j, k)$ as:

$$p_t(i, j, k) = \sigma(z_t(i, j, k)) = \frac{1}{1 + e^{-z_t(i,j,k)}}, \tag{2}$$

representing the model's confidence that voxel $(i, j, k)$ belongs to the foreground.

Uncertainty is quantified by the proximity of $p_t(i, j, k)$ to a confidence threshold $\tau \in (0, 0.5]$, which measures the distance from the decision boundary ($p = 0.5$). Voxels near this boundary correspond to ambiguous predictions:

$$\mathcal{U}_t = \{(i, j, k) \mid |p_t(i, j, k) - 0.5| < \tau\}. \tag{3}$$

In practice, $\tau$ is empirically set to 0.1, indicating that voxels with probabilities between 0.4 and 0.6 are treated as uncertain.

To incorporate user feedback, corrective clicks are simulated as $\mathbf{c}_n = (x_n, y_n, z_n)$ with associated labels $l_n \in \{0, 1\}$ (positive for inclusion, negative for exclusion). The corresponding error region is defined by the mismatch between the current prediction $\hat{y}_t$ and user intent:

$$\mathcal{E}_t = \bigcup_n \{(i, j, k) \mid \|\mathbf{x} - \mathbf{c}_n\|_2 \leq r \text{ and } \hat{y}_t(i, j, k) \neq l_n\}, \tag{4}$$

where $r$ denotes the local radius for sampling around each click. The union of uncertainty and error regions forms the candidate correction set.

Both $\mathcal{U}_t$ and $\mathcal{E}_t$ are ranked by confidence and truncated or zero-padded to a fixed number $K$ (default $K = 200$) to balance GPU memory usage. Algorithm 1 summarizes the procedure.

---

**Algorithm 1** Uncertainty- and Error-Guided Sampling at Iteration $t$

---

**Input:** Probability map $\mathbf{P}_t$, binary mask $\hat{y}_t$, user clicks $\{\mathbf{c}_n, l_n\}$, threshold $\tau$, radius $r$, limit $K$
**Output:** Sets $\mathcal{U}_t, \mathcal{E}_t$
1: $\mathcal{U}_t \leftarrow \emptyset, \mathcal{E}_t \leftarrow \emptyset$
2: **for** each voxel $(i, j, k)$ **do**
3:    **if** $|p_t(i, j, k) - 0.5| < \tau$ **then**
4:        $\mathcal{U}_t \leftarrow \mathcal{U}_t \cup \{(i, j, k, |p_t(i, j, k) - 0.5|)\}$
5:    **end if**
6: **end for**
7: **for** each click $\mathbf{c}_n$ with label $l_n$ **do**
8:    **for** voxels $\mathbf{x}$ within radius $r$ of $\mathbf{c}_n$ **do**
9:        **if** $\hat{y}_t(\mathbf{x}) \neq l_n$ **then**
10:            $\mathcal{E}_t \leftarrow \mathcal{E}_t \cup \{\mathbf{x}\}$
11:        **end if**
12:    **end for**
13: **end for**
14: Rank $\mathcal{U}_t$ by $|p_t - 0.5|$ and select top $K$ entries
15: Randomly sample or pad $\mathcal{E}_t$ to size $K$

---

### 2.3   State Tensor Construction

After sampling uncertainty points $\mathcal{U}_t$ and error points $\mathcal{E}_t$, we construct a structured state tensor for GRU-based sequential correction. The process for constructing the sequential state tensor is illustrated in Fig. 2. It consists of three horizontally arranged modules: the top module extracts **Uncertain Region Information**, the middle module extracts **Most Likely Error Region Information**, and the bottom module constructs the combined **State Tensor**.

Each sampled voxel is represented by an 8-dimensional feature vector that encodes spatial, probabilistic, and interaction information. Specifically, the first three dimensions correspond to the voxel's normalized spatial coordinates

$(i/D, j/H, k/W)$, capturing its relative position within the volumetric input. The fourth dimension stores either the model's predicted probability $p_t(i, j, k)$ for uncertainty points or the user-provided label $y_{\text{user}} \in \{0, 1\}$ for error points. The fifth dimension is a binary validity flag $m \in \{0, 1\}$, which distinguishes real voxels from padding entries introduced to maintain a fixed tensor size. The final three dimensions encode the coordinates of the initial user click $(x_0, y_0, z_0)$, providing global contextual information that is repeated for all sampled points.

Concatenating these components gives the final 8-dimensional feature vector for each voxel. Features from uncertainty and error points are separately organized as tensors:

$$\mathbf{F}_t^{\text{unc}} \in R^{K \times 8}, \tag{5}$$

$$\mathbf{F}_t^{\text{err}} \in R^{K \times 8}, \tag{6}$$

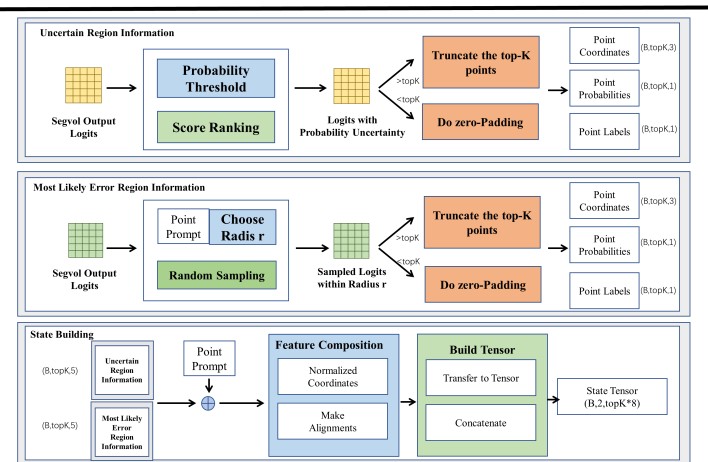

Fig. 2: Workflow for constructing the sequential state tensor from interaction points. The top horizontal container illustrates the extraction of **Uncertain Region Information**, where voxels with probabilities near the segmentation threshold ($\tau$) are identified. The middle container depicts the extraction of **Most Likely Error Region Information**, highlighting regions around user-provided corrective clicks where the current prediction disagrees with the user label. The bottom container shows the **State Construction** process, where features from both uncertainty and error regions are combined into a structured tensor suitable for GRU-based sequential correction.

and then vertically concatenated to form the combined input:

$$\mathbf{X}_t = [\mathbf{F}_t^{\mathrm{unc}}; \mathbf{F}_t^{\mathrm{err}}] \in R^{2K \times 8}. \tag{7}$$

For geometric alignment during refinement, the original spatial coordinates are stored separately as

$$\mathbf{C}_t = [\mathrm{coordinates}(\mathcal{U}_t); \mathrm{coordinates}(\mathcal{E}_t)] \in R^{2K \times 3}. \tag{8}$$

This tensor $\mathbf{X}_t$ together with $\mathbf{C}_t$ is passed to the GRU network at step $t$, allowing the model to incorporate both local uncertainty/error cues and global contextual information while maintaining a fixed-size, memory-efficient representation.

## 2.4   GRU-Based Sequential Correction

To capture temporal dependencies across interactions, we employ a Gated Recurrent Unit (GRU) network parameterized by $\theta$. At each iteration $t$, it processes the current feature tensor $\mathbf{X}_t$ and hidden state $\mathbf{h}_{t-1}$ to output probability and coordinate adjustments:

$$\Delta p_t, \Delta \mathbf{C}_t = \mathrm{GRU}_\theta(\mathbf{X}_t, \mathbf{h}_{t-1}), \tag{9}$$

where $\Delta p_t \in R^{2K}$ denotes the predicted probability offsets for the selected voxels, and $\Delta \mathbf{C}_t \in R^{2K \times 3}$ represents the coordinate refinements. The updated hidden state $\mathbf{h}_t$ encodes accumulated interaction history.

### 2.5   Iterative Refinement

After each iteration, voxel probabilities at sampled coordinates $\mathcal{C}_t$ are updated by:

$$p_{t+1}(i,j,k) = p_t(i,j,k) + \Delta p_t^{(n)}, \quad \text{for } (i,j,k) \in \mathcal{C}_t, \tag{10}$$

followed by mask binarization:

$$\hat{y}_{t+1} = I(p_{t+1} > 0.5). \tag{11}$$

This process continues for $T$ interaction rounds or until convergence. In contrast to previous memory-heavy methods, only a sparse set of uncertain or erroneous voxels is refined at each step, significantly improving computational efficiency while maintaining segmentation accuracy.

## 3   Experiments

### 3.1   Dataset and evaluation metrics

The development set is an extension of the CVPR 2024 MedSAM on Laptop Challenge [7], including more 3D cases from public datasets[1] and covering commonly used 3D modalities, such as Computed Tomography (CT), Magnetic Resonance Imaging (MRI), Positron Emission Tomography (PET), Ultrasound, and Microscopy images. The hidden testing set is created by a community effort where all the cases are unpublished. The annotations are either provided by the data contributors or annotated by the challenge organizer with 3D Slicer [4] and MedSAM2 [8]. In addition to using all training cases, the challenge contains a coreset track, where participants can select 10% of the total training cases for model development.

For each iterative segmentation, the evaluation metrics include Dice Similarity Coefficient (DSC) and Normalized Surface Distance (NSD) to evaluate the segmentation region overlap and boundary distance, respectively. The final metrics used for the ranking are:

*DSC_AUC and NSD_AUC Scores* The **AUC (Area Under the Curve)** for DSC and NSD measures cumulative improvement across interactions. This metric quantifies performance gains over the five click predictions, offering a comprehensive view of segmentation refinement. Notably, the AUC is computed solely based on click predictions, excluding the initial bounding box prediction (which is optional).

---

[1] A complete list is available at https://medsam-datasetlist.github.io/

*Final DSC and NSD Scores* These scores reflect the model's segmentation performance after all refinements, representing the ultimate accuracy achieved. In addition, the algorithm runtime will be limited to 90 seconds per class. Exceeding this limit will lead to all DSC and NSD metrics being set to 0 for that test case.

### 3.2   Implementation details

**Preprocessing** Following the practice in MedSAM [6], all images were processed to npz format with an intensity range of $[0, 255]$. Specifically, for CT images, we initially normalized the Hounsfield units using typical window width and level values: soft tissues (W:400, L:40), lung (W:1500, L:-160), brain (W:80, L:40), and bone (W:1800, L:400). Subsequently, the intensity values were rescaled to the range of $[0, 255]$. For other images, we clipped the intensity values to the range between the 0.5th and 99.5th percentiles before rescaling them to the range of $[0, 255]$. If the original intensity range is already in $[0, 255]$, no preprocessing was applied.

Following the practice in SegVol-for-SegFM,to enhance foreground contrast, we implemented dynamic intensity normalization through the `ForegroundNorm` method, which first identifies foreground voxels using an adaptive mean-intensity threshold. The intensities are then clipped to the 0.05th and 99.95th percentiles of the foreground distribution before standardization, reducing sensitivity to extreme outliers while preserving tissue contrast. For multi-class segmentation tasks, ground truth masks are automatically decomposed into binary channels for each non-background category, with explicit validation of spatial alignment between image and mask dimensions.

For memory-efficient processing of high-resolution volumetric data, we utilize sparse matrix storage (NPZ format) for ground truth masks during loading, converting to dense arrays only when necessary for computational operations. The preprocessing pipeline supports both file-based and direct array inputs, allowing flexible integration with different data sources while maintaining consistent internal representations. All spatial transformations, including resizing and cropping, are applied after initial intensity normalization to minimize intermediate memory allocation. During training, stochastic patch extraction is performed through a weighted augmentation strategy that balances computational cost and diversity, prioritizing positive-negative sampling crops where appropriate.

**Environment settings** The development environments and requirements are presented in Table 1.

**Training protocols** Building upon the successful practices established in SegVol [1], we have developed an enhanced training methodology with optimized data augmentation and sampling strategies.

Table 1: Development environments and requirements. (mandatory table)

| System | Ubuntu 22.04.2 LTS |
|---|---|
| CPU | Intel(R) Core(TM) i9-10920X CPU @ 3.50GHz |
| RAM | $16\times32$GB; 3200MT$/s$ |
| GPU (number and type) | Two NVIDIA GeForce RTX 4090 24G |
| CUDA version | 12.4 |
| Programming language | Python 3.10.16 |
| Deep learning framework | torch 2.7.0+cu126 torchvision 0.14.1+cu117 |

**Data Augmentation:** The training pipeline incorporates several spatial and intensity transformations to improve model generalization. Spatial augmentations include random flipping along all three axes (sagittal, coronal, and axial planes) with a probability of 0.2 for each orientation. For volumetric data, we employ a mixed strategy of either full-volume resizing or patch extraction through positive-negative sampling, with sampling weights favoring the latter (3:1 ratio). The patch extraction uses `RandCropByPosNegLabeld` with 3 positive samples for every negative sample, ensuring adequate representation of both foreground and background regions. Intensity augmentations consist of random scaling (factor range $\pm20\%$) and shifting ($\pm20\%$ of intensity range), each applied with 20% probability.

**Data Sampling Strategy:** We implement a dynamic sampling approach that adapts to the multi-class nature of segmentation tasks. During batch construction, the system automatically balances class representation by: 1) Preserving all available foreground classes in each sample through binary mask decomposition 2) Applying foreground cropping to concentrate computation on relevant regions 3) Using sparse storage formats for ground truth masks during loading to enable memory-efficient handling of large 3D volumes 4) Supporting both whole-volume processing and patch-based training, with the latter preferentially sampling regions containing segmentation targets when available. The sampling weights for patch extraction favor positive regions (3:1 positive-to-negative ratio) to address class imbalance while maintaining context. For inference, we process full volumes with optional overlapping sliding window when necessary for large scans.

**Interactive Optimization Strategy** We present a two-stage training strategy that harmonizes global shape consistency with local detail refinement to achieve an optimal balance between interaction efficiency and segmentation accuracy in medical image analysis. The framework employs box prompt losses in the first stage to establish topological validity through global similarity metrics, followed by a second stage that combines both box and point prompt losses to enable precise boundary refinement using distance metrics, thereby preventing overfitting to initial prompts while ensuring robust convergence.

To accurately reflect clinical workflows where users typically perform limited refinements, we implement a weighted training protocol that prioritizes 2-3 GRU-based refinement iterations as the most frequent scenario, with 1 and 4 iterations occurring less frequently, and 0 or 5 iterations representing rare cases. This distribution ensures the model learns predominant interaction patterns while maintaining adaptability to extreme cases. The optimization process incorporates sequential refinement mechanisms that maintain memory states between iterations, coupled with a composite loss function that simultaneously optimizes global structural similarity, voxel-wise prediction accuracy, and consistency between initial and refined outputs. The conditional execution architecture enables differentiated processing of multimodal interaction prompts while ensuring stable gradient flow throughout the refinement cascade, providing both the efficiency required for clinical practice and the flexibility needed for complex segmentation tasks.

Table 2: Training protocols. (mandatory table) Please fill out all rows

| Pre-trained Model | SegVol (for SegFM) |
|---|---|
| Batch size | 4 |
| Patch size | $256{\times}256{\times}32$ |
| Total epochs | 25 |
| Optimizer | AdamW |
| Initial learning rate (lr) | 1e-5 |
| Lr decay schedule | None |
| Training time | 2 days 9 hours |
| Loss function | Dice loss + BCE loss |
| Number of model parameters | 295.35M[2] |
| Number of flops | 264.18G[3] |

## 4    Results

### 4.1    Analysis of the Iterative Refinement Mechanism

Our framework is fundamentally designed to enhance segmentation accuracy through a progressive, multi-stage correction process. To validate the effectiveness of this core mechanism, we first conducted an analysis focused on the performance gains achieved at each step of the iterative refinement. This evaluation was performed on a representative 5% subset of the validation set to enable rapid prototyping and assessment of the iterative dynamics.

As detailed in Table 3, the model demonstrates a clear and consistent pattern of improvement with each correction iteration. Starting from an initial segmentation generated by the first prompt, which achieved a Dice Similarity Coefficient

(DSC) of 0.661, the model's performance steadily increases. The first correction step yields the most substantial improvement, boosting the DSC to 0.669 relative gain of 1.18%. Subsequent corrections continue to fine-tune the segmentation boundaries, with the second and third iterations contributing further incremental gains of +0.17% and +0.11%, respectively. After three refinement stages, the final DSC score stabilized at 0.671, achieving a total absolute improvement of approximately +0.01 over the initial baseline.

This iterative enhancement pattern confirms that our proposed correction module effectively identifies and rectifies initial segmentation errors. The diminishing returns in later stages suggest that the model converges towards a more accurate solution. Having established the internal validity and efficacy of our iterative approach, we then proceeded to conduct a comprehensive comparative evaluation against state-of-the-art methods across the full spectrum of imaging modalities to benchmark its overall performance.

Table 3: Performance Improvement Through Iterative Refinement (based on 5% validation set)

| Iteration | DSC Value | Absolute Improvement | Relative Improvement (%) |
| --- | --- | --- | --- |
| Initial Prompt | 0.661081 | - | - |
| First Correction | 0.668906 | +0.00783 | +1.18% |
| Second Correction | 0.670053 | +0.00115 | +0.17% |
| Third Correction | 0.670816 | +0.00076 | +0.11% |

## 4.2 Comparative Evaluation on Multi-Modal Datasets

We conducted a comprehensive comparative evaluation to benchmark our framework against several state-of-the-art (SOTA) interactive segmentation methods, including SAM-Med3D, VISTA3D, SegVol and nnInteractive. The evaluation was performed across a diverse range of five medical imaging modalities: Computed Tomography (CT), Magnetic Resonance Imaging (MRI), Microscopy, Positron Emission Tomography (PET), and Ultrasound (US). The objective was to assess the generalization capability and overall performance of our method in various clinical and research scenarios.

The quantitative results on validation sets, summarized in Table 4, reveal a highly competitive performance landscape. Our method consistently places among the top performers, demonstrating particular strength in clinically prevalent modalities. Notably, it achieves state-of-the-art results in MRI, securing the highest scores in both DSC AUC (2.960) and NSD AUC (3.449). Furthermore, our framework delivers top-tier performance in CT and US modalities, outperforming most competing approaches and underscoring its robustness and versatility.

This quantitative superiority is directly corroborated by a qualitative analysis of the segmentation results. For modalities with well-defined structures, such as

CT scans with clear anatomical boundaries, our method consistently produces high-fidelity segmentations. As shown in Figure 3, our approach (l) successfully renders a smooth and coherent 3D model that is visibly more complete than that of SegVol (i), directly explaining its improved accuracy reported in Table 4.

The advantages of our framework are even more pronounced in MRI datasets, which are characterized by high soft-tissue contrast. This environment is particularly well-suited for our iterative refinement mechanism. As illustrated in Figure 5, our method (l) excels at producing a remarkably complete and detailed 3D segmentation, overcoming the challenges of ambiguous tissue boundaries that lead to fragmented results in other models like SegVol (i). This superior capability to handle complex soft-tissue anatomy is the primary driver behind our state-of-the-art quantitative scores in the MRI category.

Conversely, the evaluation highlights the shared challenges posed by modalities with inherent physical limitations. The primary failure cases for all methods occur in PET images with low spatial resolution, where the limited anatomical detail and inherent blurriness hinder the accurate delineation of fine structures (Figure 6). Similarly, while our method performs well quantitatively on Ultrasound data, the presence of speckle noise and acoustic shadowing artifacts remains a challenge for achieving perfect boundary precision (Figure 7), an area for future improvement for all segmentation algorithms.

In summary, the comprehensive benchmark validates our framework's position as a robust and high-performing interactive segmentation tool. The analysis confirms that our iterative refinement strategy is particularly effective in complex clinical modalities like MRI and CT, while soberly acknowledging the shared challenges faced by all current methods in lower-quality imaging environments.

This strong quantitative performance is directly corroborated by a qualitative analysis of the segmentation results, which reveals the underlying reasons for our method's success and limitations in different imaging environments.

For modalities with well-defined structures, such as CT scans with clear anatomical boundaries, our method consistently produces high-fidelity segmentations. As shown in Figure 3, our approach (l) successfully renders a smooth and coherent 3D model. Compared to SegVol (i), which misses several smaller anatomical components, our method's result is visibly more complete, which directly explains its improved segmentation accuracy reported in Table 4.

The advantages of our framework are even more pronounced in MRI datasets, which are characterized by high soft-tissue contrast. This environment is particularly well-suited for our iterative refinement mechanism. As illustrated in Figure 5, our method (l) excels at producing a remarkably complete and detailed 3D reconstruction, overcoming the challenges of ambiguous tissue boundaries that lead to fragmented results in other models like SegVol (i). This superior capability to handle complex soft-tissue anatomy is the primary driver behind our state-of-the-art quantitative scores in the MRI category.

Conversely, the model's performance, along with that of all competing methods, is challenged by modalities with inherent physical limitations. The primary failure cases occur in: PET images with low spatial resolution: As seen in Fig-

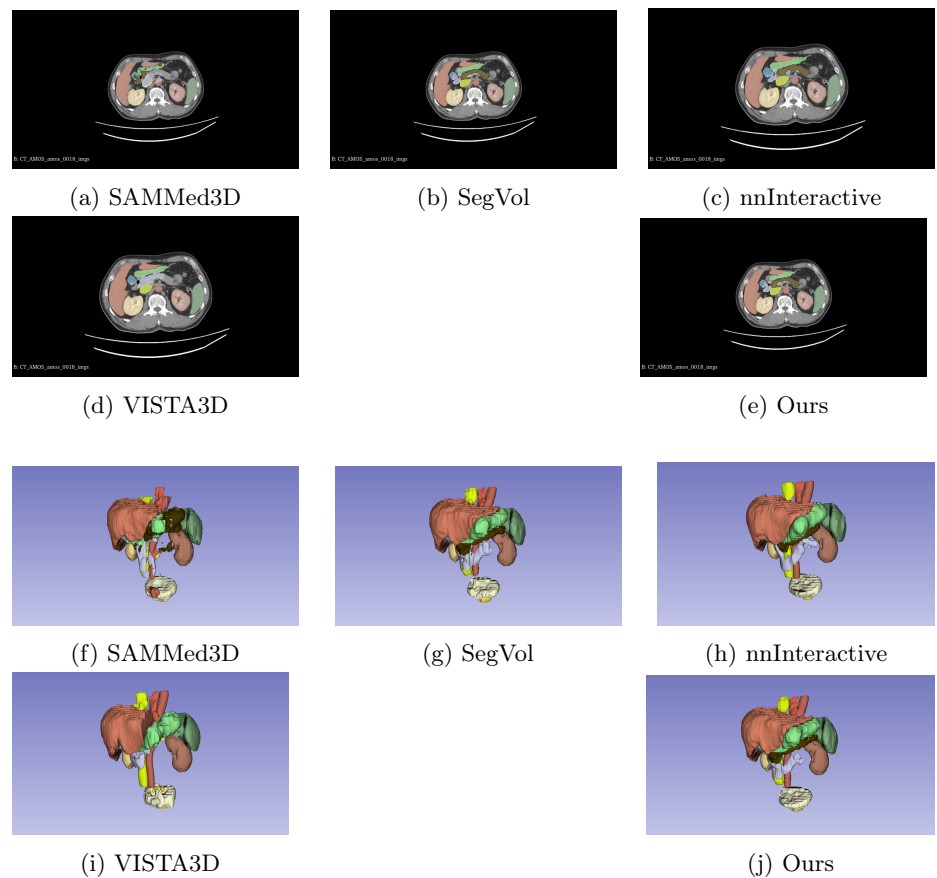

Fig. 3: Qualitative comparison of different segmentation methods on an abdominal CT scan. The first row (a-e) shows the 2D cross-sectional segmentation results from SAMMed3D, SegVol, nnInteractive, VISTA3D, and our method, respectively. The second row (f-j) shows the corresponding 3D rendered visualizations.

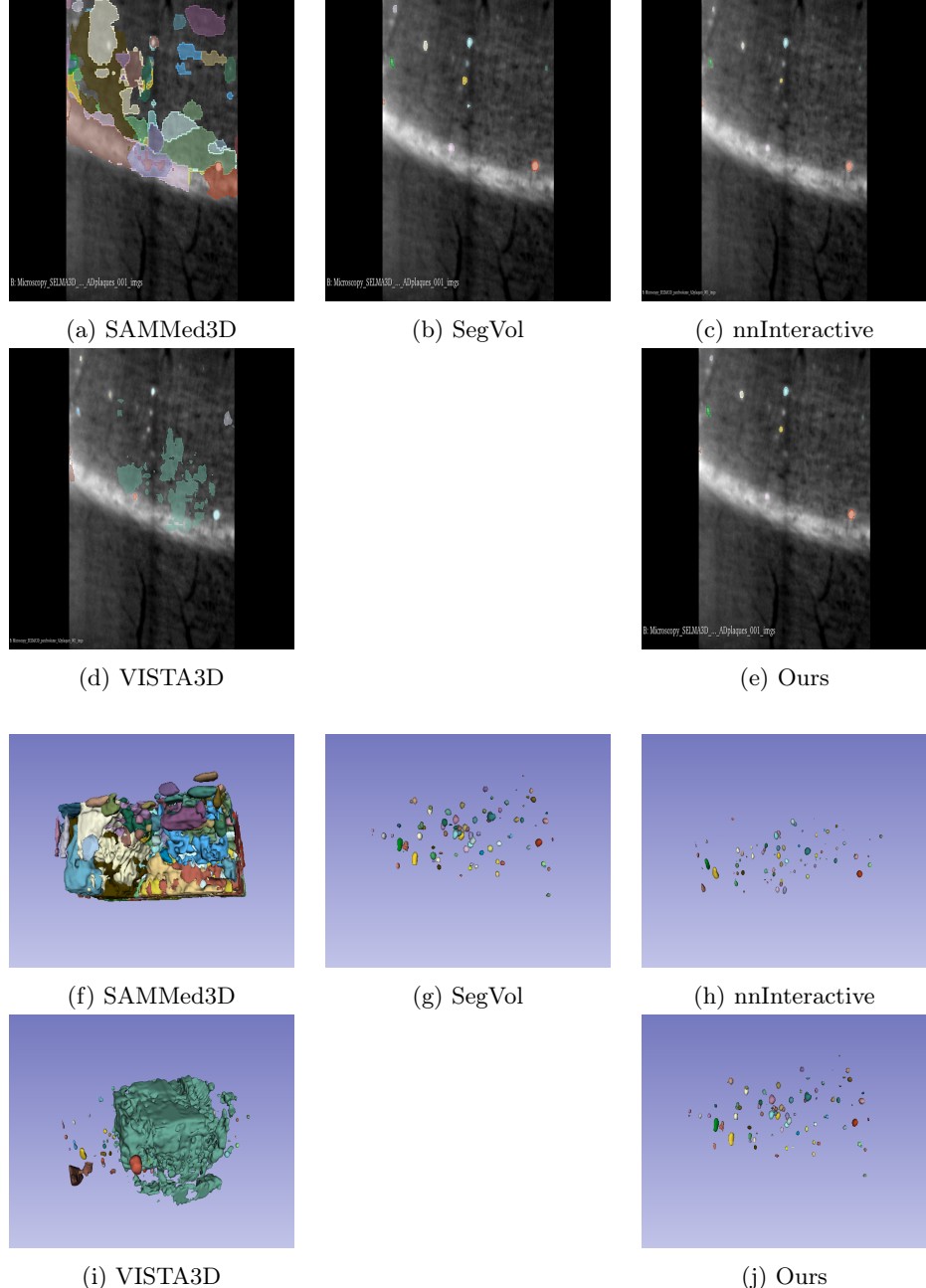

Fig. 4: Qualitative comparison of different segmentation methods on an abdominal Microscopy scan. The first row (a-e) shows the 2D cross-sectional segmentation results from SAMMed3D, SegVol, nnInteractive, VISTA3D, and our method, respectively. The second row (f-j) shows the corresponding 3D rendered visualizations.

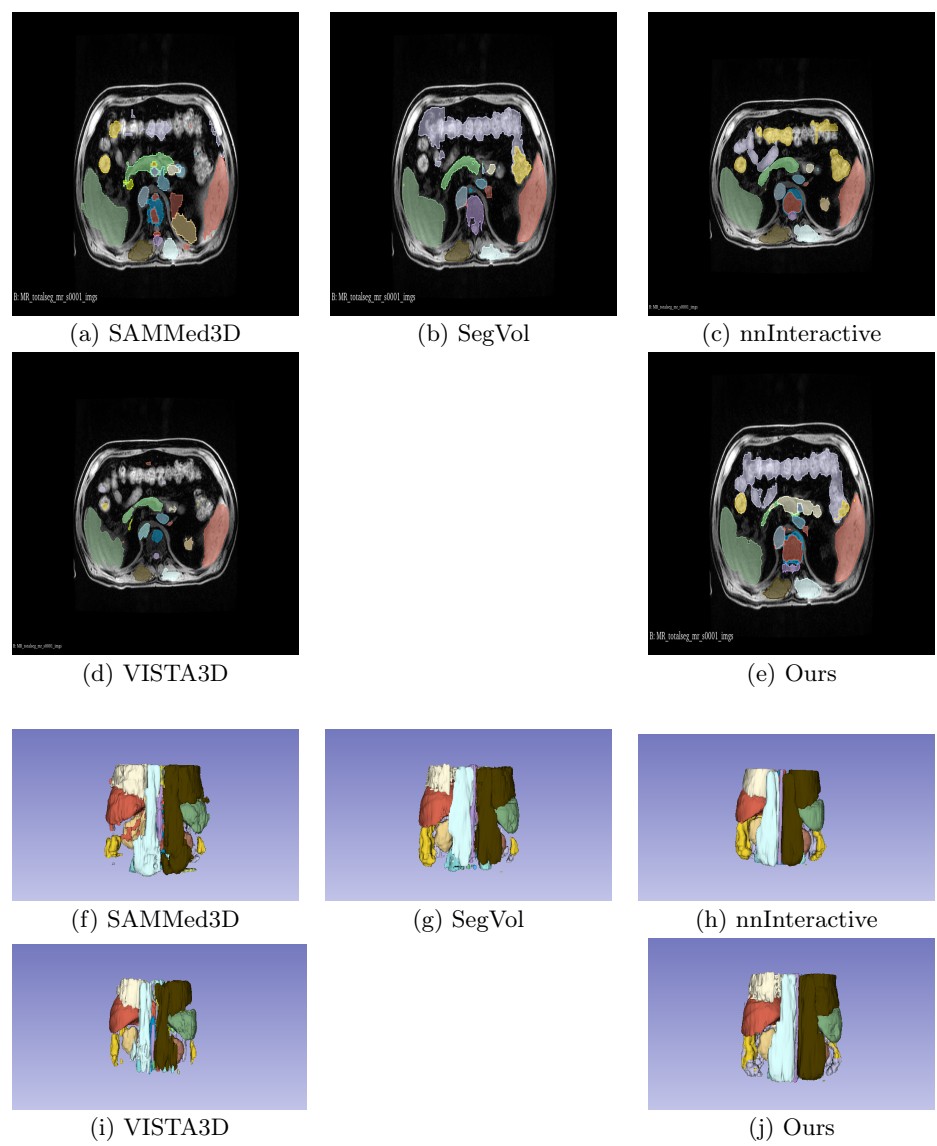

Fig. 5: Qualitative comparison of different segmentation methods on an abdominal MRI scan. The first row (a-e) shows the 2D cross-sectional segmentation results from SAMMed3D, SegVol, nnInteractive, VISTA3D, and our method, respectively. The second row (f-j) shows the corresponding 3D rendered visualizations.

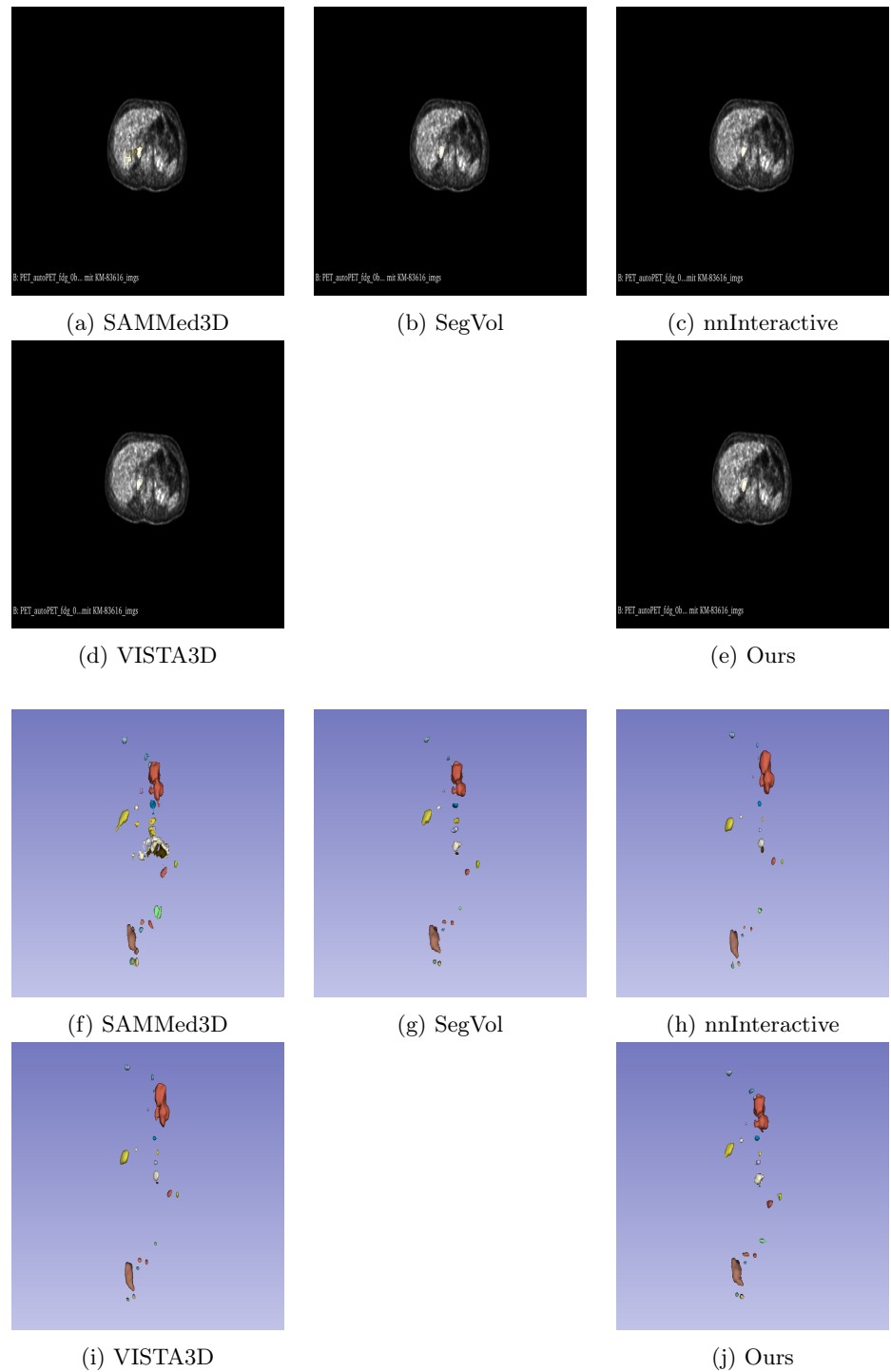

(a) SAMMed3D            (b) SegVol            (c) nnInteractive

(d) VISTA3D                          (e) Ours

(f) SAMMed3D            (g) SegVol            (h) nnInteractive

(i) VISTA3D                          (j) Ours

Fig. 6: Qualitative comparison of different segmentation methods on an abdominal PET scan. The first row (a-e) shows the 2D cross-sectional segmentation results from SAMMed3D, SegVol, nnInteractive, VISTA3D, and our method, respectively. The second row (f-j) shows the corresponding 3D rendered visualizations.

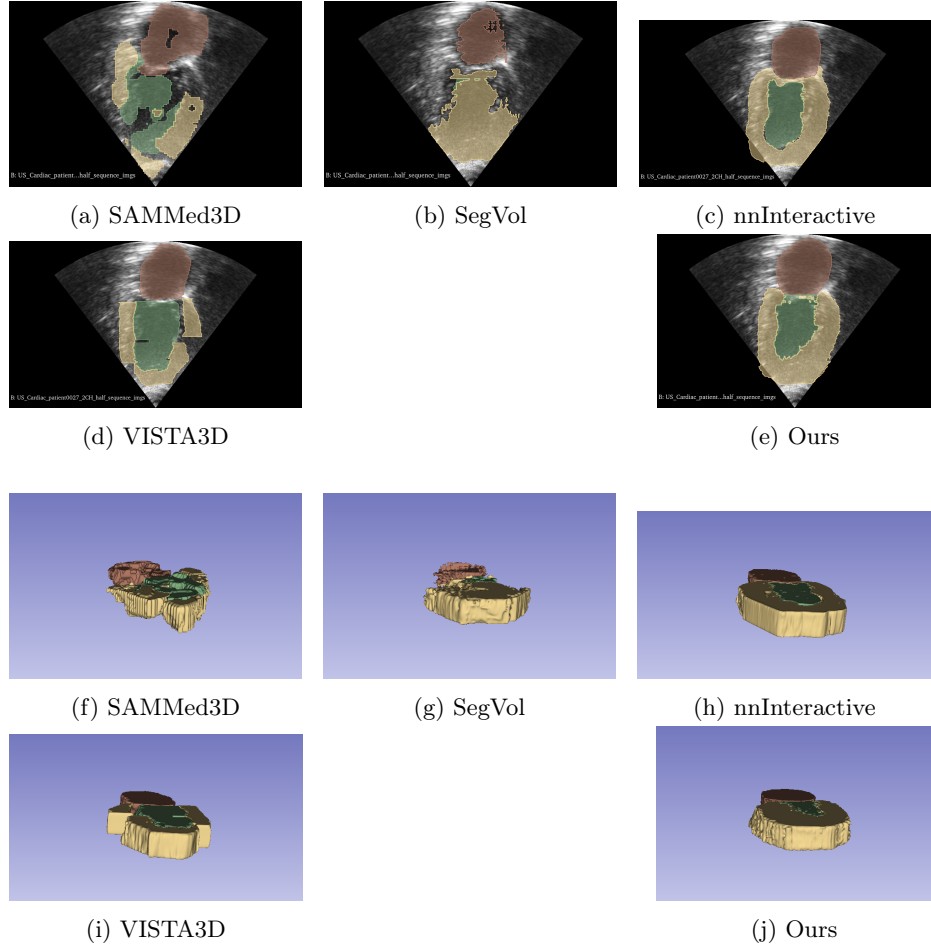

(a) SAMMed3D          (b) SegVol          (c) nnInteractive

(d) VISTA3D                              (e) Ours

(f) SAMMed3D          (g) SegVol          (h) nnInteractive

(i) VISTA3D                              (j) Ours

Fig. 7: Qualitative comparison of different segmentation methods on an abdominal US scan. The first row (a-e) shows the 2D cross-sectional segmentation results from SAMMed3D, SegVol, nnInteractive, VISTA3D, and our method, respectively. The second row (f-j) shows the corresponding 3D rendered visualizations.

Table 4: Comparison of segmentation methods on validation sets.

| Modality | Method | DSC_AUC | NSD_AUC | DSC_Final | NSD_Final |
|---|---|---|---|---|---|
| CT | SAM-Med3D | 2.2408 | 2.2213 | 0.5590 | 0.5558 |
| | VISTA3D | 3.1689 | 3.2652 | 0.8041 | 0.8344 |
| | SegVol | 2.9809 | 3.1235 | 0.7452 | 0.7809 |
| | nnInteractive | 3.4337 | 3.5743 | 0.8764 | 0.9165 |
| | Our | 3.155 | 3.383 | 0.789 | 0.846 |
| MR | SAM-Med3D | 1.5222 | 1.5226 | 0.3903 | 0.3964 |
| | VISTA3D | 2.5895 | 2.9683 | 0.6545 | 0.7493 |
| | SegVol | 2.6719 | 3.1535 | 0.6680 | 0.7884 |
| | nnInteractive | 2.6975 | 3.0292 | 0.7302 | 0.8227 |
| | Our | 2.960 | 3.449 | 0.743 | 0.879 |
| Microscopy | SAM-Med3D | 0.1163 | 0.0000 | 0.0291 | 0.0000 |
| | VISTA3D | 2.1196 | 3.2259 | 0.5478 | 0.8243 |
| | SegVol | 1.6846 | 2.9716 | 0.4211 | 0.7429 |
| | nnInteractive | 2.3311 | 3.1109 | 0.5943 | 0.7890 |
| | Our | 2.108 | 3.107 | 0.526 | 0.776 |
| PET | SAM-Med3D | 2.1304 | 1.7250 | 0.5344 | 0.4560 |
| | VISTA3D | 2.6398 | 2.3998 | 0.6779 | 0.6227 |
| | SegVol | 2.9683 | 2.8563 | 0.7421 | 0.7141 |
| | nnInteractive | 3.1877 | 3.0722 | 0.8156 | 0.7915 |
| | Our | 2.923 | 2.805 | 0.730 | 0.701 |
| US | SAM-Med3D | 1.4347 | 1.9176 | 0.4102 | 0.5435 |
| | VISTA3D | 2.8655 | 2.8441 | 0.8105 | 0.8079 |
| | SegVol | 1.2438 | 1.8045 | 0.3109 | 0.4511 |
| | nnInteractive | 3.3481 | 3.3236 | 0.8547 | 0.8494 |
| | Our | 3.038 | 3.146 | 0.759 | 0.786 |

[*] All methods exclude failed cases from metrics. Empty segmentations receive score 0 in official evaluation.

Table 5: Performance comparison across modalities and methods on test sets.

| Modality | Method | DSC_AUC | NSD_AUC | DSC_Final | NSD_Final |
|---|---|---|---|---|---|
| CT | SAM-Med3D | 2.1937 | 1.7846 | 0.5711 | 0.4672 |
| | VISTA3D | 2.3482 | 2.1062 | 0.6198 | 0.5616 |
| | SegVol | 2.4358 | 2.3213 | 0.6089 | 0.5803 |
| | nnInteractive | 3.1831 | 3.1286 | 0.8342 | 0.8355 |
| | our | 2.3187 | 2.2183 | 0.5824 | 0.5570 |
| MR | SAM-Med3D | 2.1064 | 2.0427 | 0.5317 | 0.5169 |
| | VISTA3D | 2.4891 | 2.5825 | 0.6516 | 0.6859 |
| | SegVol | 2.8377 | 3.1261 | 0.7094 | 0.7815 |
| | nnInteractive | 3.3866 | 3.6611 | 0.8680 | 0.9416 |
| | our | 2.7898 | 3.0283 | 0.6984 | 0.7576 |
| Microscopy | SAM-Med3D | 0.3115 | 0.1726 | 0.0778 | 0.0431 |
| | VISTA3D | 2.4526 | 3.4035 | 0.6231 | 0.8528 |
| | SegVol | 2.9603 | 3.9472 | 0.7401 | 0.9868 |
| | nnInteractive | 3.4580 | 3.9895 | 0.8743 | 0.9980 |
| | our | 3.0123 | 3.9737 | 0.7551 | 0.9942 |
| PET | SAM-Med3D | 1.3004 | 0.7297 | 0.3285 | 0.1844 |
| | VISTA3D | 1.8687 | 1.3919 | 0.4688 | 0.3523 |
| | SegVol | 2.9844 | 2.5108 | 0.7461 | 0.6277 |
| | nnInteractive | 3.2230 | 3.0753 | 0.8170 | 0.7854 |
| | our | 2.9323 | 2.4794 | 0.7329 | 0.6202 |
| US | SAM-Med3D | 0.8313 | 0.7004 | 0.2078 | 0.1751 |
| | VISTA3D | 0.9072 | 1.2257 | 0.2953 | 0.4789 |
| | SegVol | 0.9429 | 1.4435 | 0.2357 | 0.3609 |
| | nnInteractive | 2.4088 | 3.0407 | 0.7073 | 0.8886 |
| | our | 0.9949 | 1.5788 | 0.2652 | 0.4112 |

ure 6, the limited anatomical detail and inherent blurriness of PET scans hinder the accurate delineation of fine structures, a fundamental challenge for any segmentation algorithm. Ultrasound artifacts: Despite achieving strong quantitative results, ultrasound imaging (Figure 7) remains a difficult modality. The presence of speckle noise and acoustic shadowing effects can occasionally compromise the model's robustness, leading to less precise boundaries compared to CT or MRI. The strong performance reported in Table 4 highlights our model's notable resilience to these artifacts compared to other methods, though it remains an area for future improvement. In summary, the comprehensive benchmark across five distinct modalities validates our framework's position as a robust, versatile, and high-performing interactive segmentation tool. The qualitative analysis confirms that our iterative refinement strategy is particularly effective in complex clinical modalities like MRI and CT, while soberly acknowledging the shared challenges faced by all current methods in lower-quality imaging environments.

## 5    Discussion and Limitation

Our framework redefines interactive 3D medical image segmentation by emphasizing targeted, lightweight correction based on uncertainty and error heuristics. Unlike prior works that rely on memory-aware backbone architectures or multimodal generalization strategies, our method selectively processes only those regions with high uncertainty or localized label disagreements, substantially reducing computational overhead while maintaining effective refinement. This approach enables progressive improvement in segmentation accuracy with each interaction, demonstrating diminishing yet consistent gains over successive iterations.

Despite its effectiveness, several limitations remain. First, the notion of uncertainty—voxels with probabilities near the decision threshold—is useful for identifying ambiguous regions but is not explicitly calibrated; it does not guarantee coverage of all true error regions. Similarly, the "error" set, defined by local disagreement with user-provided clicks, may overlook systematic or spatially distributed prediction errors. Consequently, the GRU corrector can only address a subset of potential errors, and comprehensive volumetric correction is not ensured.

Additional limitations include memory-constrained input sampling and the external nature of sequential modeling. At each interaction step, only a fixed number of points (e.g., 200) are sampled due to GPU constraints, limiting the model's ability to capture long-range spatial context. Moreover, the GRU operates outside the segmentation backbone, receiving low-dimensional features rather than full encoder-decoder representations. This separation restricts deeper temporal context modeling and may constrain correction potential in complex anatomical regions.

To address these limitations, several avenues for future work are envisioned. Scaling up the number of sampled points would allow richer contextual cues from both uncertainty and error distributions. Integrating the GRU directly into the segmentation backbone could enable prompt-aware, temporally consistent refinement within high-dimensional feature space. Finally, end-to-end temporal learning would allow joint optimization of segmentation and memory updates, potentially improving both convergence and overall accuracy.

## 6    Conclusion

We have presented a novel framework for interactive 3D medical image segmentation that leverages sequential, history-aware refinement driven by uncertainty and error heuristics. By focusing computation on the most informative regions, our approach achieves a lightweight yet effective correction mechanism, improving segmentation accuracy progressively over multiple interactions.

Extensive evaluation across diverse imaging modalities—CT, MRI, Microscopy, PET, and Ultrasound—demonstrates that our method consistently enhances segmentation quality, particularly in anatomically complex regions or modalities

with challenging contrast. While our current heuristics may not guarantee exhaustive coverage of all error regions, the approach provides a practical balance between efficiency, memory usage, and interactive performance.

Overall, this work underscores the value of targeted, iterative correction in interactive segmentation, offering a robust and versatile tool that can accelerate medical image annotation and support precise, clinically relevant analyses.

**Acknowledgements** We thank all the data owners for making the medical images publicly available and CodaLab [10] for hosting the challenge platform.

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
