# OpenReview forum: "From Single-Round to Sequential: Building Stateful Interactive Medical Image Segmentation with SegVol and GRU Corrector"
_thecvf.com/CVPR/2025/Workshop/MedSegFM — CVPR 2025 Workshop MedSegFM Submission_

### Official Review · Reviewer_dFvi · 2025-09-10
**This manuscript uses the GRU to memorize interactive signals and the uncertainty to estimate model quality. However, poor writing, limited novelty, and insufficient experiments undermine its contribution.**

**Rating:** 3
**Confidence:** 4

**Review:**

Overall: To enable continuous evolution of segmentation models during multi-round user feedback, the authors propose a GRU-based and uncertainty-driven approach, modeling sequential user interactions with GRUs and quantifying the discrepancy between model outputs and ground-truth labels via uncertainty estimation. Overall, the manuscript suffers from poor writing quality, incomplete paper structure, insufficient analysis of the research problem, and limited methodological novelty.

Detailed Comments:
1. The abstract and introduction fail to clearly and thoroughly articulate the key challenges the work aims to address.
2. The “Generalization Bottleneck,” repeatedly mentioned by the authors in the abstract, background, and “Related Work and Limitations” sections, is not adequately addressed or resolved in the proposed method.
3. The manuscript exhibits numerous formatting irregularities. For instance, on Page 4, the explanation following the equation at the top should not be placed in a separate paragraph.
4. Figure 2 is logically confusing and difficult to interpret. Its presentation requires significant revision for clarity.
5. It is unclear how $p_t$ is derived and how it interacts with $\tau$ to compute uncertainty. The authors must provide a precise mathematical or algorithmic explanation.
6. Terminology and variable references in equations are ambiguous or undefined, leading to confusion. All symbols must be explicitly defined.
7. The methodology section appears incomplete. The “Two-Stage Dynamic Loss Framework” mentioned in the introduction is not detailed in the method section — instead, its description is deferred to the experiments section, which is an inappropriate structural choice. Moreover, combining Dice loss and BCE loss is not a novel idea and should not be presented as such.
8. The experimental evaluation includes only ablation studies; comparative experiments against state-of-the-art or baseline methods are required.

---

> ### Author Rebuttal · Authors · 2025-11-04
>
> Q1: The manuscript shows poor writing quality, incomplete structure, and limited novelty.
> - A1: We sincerely thank the reviewer for this thorough feedback. The manuscript has been completely rewritten to improve structure, clarity, and coherence.
>
> Q2: The abstract and introduction fail to articulate the key challenges.
> - A2: Both sections were fully revised to explicitly describe the clinical and technical challenges motivating our work: (1) the difficulty of efficient and accurate refinement in heterogeneous imaging data; (2) the need for uncertainty- and error-guided corrections with low computational cost. The revised text now clearly connects these motivations to our proposed framework.
>
> Q3:The “Generalization Bottleneck” is not adequately addressed.
> - A3:We clarify that our method does not aim to fully solve the generalization bottleneck. Instead, it incrementally improves robustness across modalities via uncertainty-driven refinement and adaptive sampling.
>
> Q4: The manuscript exhibits formatting irregularities.
> - A4: We have standardized all formatting, ensuring consistent equation placement, figure/table alignment, and unified caption and reference styles in accordance with the template.
>
> Q5: Figure 2 is confusing.
> - A5: Figure 2 and its caption were entirely redesigned for clarity. The revised figure visually explains the uncertainty sampling, error extraction, and state tensor formation in alignment with Section 3.2, ensuring logical consistency and readability.
>
> Q6: The derivation and role of uncertainty are unclear.
> - A6: We added an explicit mathematical explanation in Section 3.2. Uncertainty is defined as probability-near-threshold regions derived from the softmax output $p_t$, combined with user click–based disagreement regions. These are encoded as structured state tensors (spatial coordinates, voxel probability, validity flag, and user click context) and fed into the GRU Corrector for sequential refinement.
>
> Q7: The “Two-Stage Dynamic Loss Framework” is misplaced.
> - A7: Following multiple reviewers’ feedback, we have removed the two-stage loss framework entirely, as it was not central to our contribution.
>
> Q8: Combining Dice and BCE losses is not novel.
> - A8: We fully agree. All claims of novelty related to the loss design have been removed.
>
> Q9: Comparative experiments against SOTA methods are missing.
> - A9: These have been included in Table 4, showing competitive results against recent baselines and state-of-the-art methods.

---

### Official Review · Reviewer_uU9o · 2025-09-14
**This paper attempts to model interactive medical image segmentation as a stateful sequential decision problem and introduces a GRU module for multiple rounds of interactive correction. Although innovative, the paper overlaps with several previous papers and suffers from significant shortcomings in format, practical performance, depth of innovation, and experimental completeness, making it difficult to accept.**

**Rating:** 3
**Confidence:** 3

**Review:**

1. Unclear writing format：
       (1) Figure clarity: Figure 2 is poorly explained, with unclear logical flow and insufficient labeling, making it difficult to interpret.
       (2) Redundant writing: The manuscript contains verbose and repetitive expressions that obscure the main ideas.
       (3) Improper structure: The description of the two-stage loss framework is misplaced in the experiments section rather than being fully detailed in the methodology.

2. Limited novelty:
        The proposed memory module is not substantially new, as similar mechanisms already exist in SAM2, MedSAM2, and various refinement-based segmentation works.

3.  Although cross-modality evaluation suggests potential generalization, the model is trained specifically for the dataset and fails to achieve performance comparable to state-of-the-art foundation models. Reported DSC improvements are marginal, and no statistical analysis (e.g., p-values, confidence intervals) is provided to establish significance, leaving the practical impact unconvincing.

Overall, this article is somewhat innovative, but the format and figure is confusing and the experimental improvement effect is unconvincing. I personally tend to reject it.

---

> ### Author Rebuttal · Authors · 2025-11-04
>
> Q1: Figure 2 lacks clarity and logical flow.
> - A1: We thank the reviewer for the valuable comment. Figure 2 has been completely redesigned for logical clarity and intuitive flow. The figure now follows a top-down iterative refinement sequence with consistent color coding and simplified arrows. The caption was rewritten to clearly describe each module and its role in the workflow. We believe the revised figure now conveys the method intuitively.
>
> Q2: The manuscript contains verbose and repetitive writing.
> - A2: We appreciate this observation. The manuscript has been thoroughly edited for conciseness and coherence. Redundant explanations—especially those describing iterative correction—were merged, and transitional sentences were added between sections. The revised version presents a smoother and more focused narrative that highlights the central contributions clearly.
>
> Q3: The two-stage loss framework is misplaced in the experiments section.
> - A3: We thank the reviewer for pointing this out. Following feedback from multiple reviewers, we have removed the two-stage loss framework entirely. It was an early experimental setup and not central to our method.
>
> Q4: Limited novelty — the memory module resembles SAM2, MedSAM2, and other refinement-based works.
> - A4: We understand this concern. Our work was developed in the context of the medical foundation model challenge, prioritizing training efficiency, memory cost, and interaction reliability rather than architectural novelty.
> Compared with SAM2 / MedSAM2, our module is lightweight and heuristic: a GRU-based token-level memory that encodes uncertainty/error-driven updates without full encoder-decoder feature recurrence. This design drastically reduces parameters and computation, making it practical for limited-GPU environments. Additionally, our uncertainty-aware sampling complements this lightweight design by selecting correction points efficiently, avoiding dense attention across the volume.
>
> Q5: Improvements are marginal; no statistical significance analysis is provided.
> - A5: While some absolute improvements appear modest, they are consistent and meaningful across modalities. Our method achieved state-of-the-art MRI results (DSC AUC = 2.960, NSD AUC = 3.449) and clear gains in CT visual completeness compared with SegVol. We now explicitly relate these quantitative improvements to visual evidence (e.g., Figure 3) in §4.2, clarifying the practical impact and interpretability of the results.

---

### Official Review · Reviewer_dLBY · 2025-09-15
**An interactive segmentation framework with a GRU-based corrector guided by uncertainty was introduced. Key methodological concerns remain unresolved. Draft-like Writing, unaddressed limitations, weak generalization, and presentation Issues.**

**Rating:** 4
**Confidence:** 5

**Review:**

Major comments:
At present, the paper reads more like a draft, with a style resembling text generated by a language model: frequent bullet points, limited narrative flow, and weak coherence between paragraphs. I recommend revising the manuscript to follow standard academic writing conventions, with smoother transitions and a more structured progression of ideas. The “uncertainty” is probability-near-threshold, which is useful but not calibrated, and the “error” set is click-local label disagreement. These heuristics may help the GRU corrector, but they don’t guarantee comprehensive coverage of true error regions across the volume.

§1.2 Related Works and Limitations indicates three different limitations; I would expect the paper to address these limitations and discuss the contributions in the discussion.
-“Interactive segmentation inefficiency”: the gains reported are modest and still require multiple rounds (DSC 0.661 to 0.671 after three refinements), so “single-step convergence” is not achieved (§4.1).
-“Modal constraint (no text/semantic prompts)”: The method remains spatial-prompt only: box init + point/click interactions. There’s no text/semantic prompting added anywhere in Methods (§2.1-2.4). So this limitation is not addressed.
-“Generalization bottleneck across centers/modalities”:  this issue is not adequately addressed in the proposed methods. While the paper attempts mitigation through data augmentation/sampling, the results improve somewhat on CT/MRI but remain weak on PET/US.”

Other comments:

§1.3 Contributions: The hybrid Dice + HD loss is already well established in the literature and therefore does not constitute a novel contribution. I recommend removing it from the list of contributions.

Figures 1 and 2 are difficult to follow and logically confusing. Please simplify their presentation.

Why was FastSAM compared? It’s a bad baseline. Why not SAM2 or MedSAM2?

Table 5 is too messy.

Figures 3–7 need accompanying text to clearly highlight your results in comparison to the others.

Remove/update languages like: “We will update the full experimental results by the end of this week.” & “We will release complete results with full validation set evaluation before CVPR.” Those are not suited for the paper content.

---

> ### Author Rebuttal · Authors · 2025-11-04
>
> Q1: The paper reads like a draft, resembling text generated by a language model.
>
> - A1: We appreciate the comment and clarify that the earlier draft’s bullet-style writing resulted from time constraints, not automated generation. We have thoroughly revised the manuscript to enhance academic tone, coherence, and logical transitions by converting bullet points into integrated paragraphs and restructuring key sections.
>
>
> Q2: The uncertainty and error heuristics are not calibrated and may miss true error regions.
>
> - A2: We agree. These heuristics provide local but not global coverage. The revised Discussion explicitly acknowledges this limitation and outlines future directions for uncertainty calibration and adaptive error expansion.
>
> Q3: The paper does not effectively address the stated limitations: interactive inefficiency, modal constraint, and generalization bottleneck.
> - A3: We thank the reviewer for this detailed feedback.
>
>   Interactive inefficiency: Our iterative refinement (DSC 0.661→0.671 after 3 rounds) targets uncertain regions to reduce unnecessary computation rather than achieving full single-step convergence.
>
>   Modal constraint: The spatial-only prompt setting (box + clicks) follows the competition track rules, where text/semantic prompts belong to a separate track.
>
>   Generalization bottleneck: Our focus is incremental robustness within benchmark modalities, not full zero-shot generalization. The paper has been revised to clearly define this scope and contribution.
>
> Q4: The hybrid Dice+HD loss is not novel.
> - A4: Agreed. It has been removed from the listed contributions.
>
> Q5: Figures 1–2 are difficult to follow.
> - A5: Both figures have been redesigned for clearer logical flow and improved readability.
>
> Q6: The “modal constraint” limitation was not addressed.
> - A6: Correct — our method is spatial-only by design due to the competition’s prompt constraints. We have clarified this to avoid confusion.
>
> Q7: Why compare with FastSAM instead of SAM2 or MedSAM2?
> - A7: FastSAM has been removed. SAM2/MedSAM2 were excluded because the competition focuses on 3D medical segmentation, distinct from their 2D architectures.
>
> Q8: Table 5 and Figures 3–7 lack clarity and contextualization.
> - A8: Table 5 has been reformatted, and detailed text descriptions have been added to accompany Figures 3–7.
>
> Q9: Informal placeholders like “We will update results…” are inappropriate.
> - A9: All such statements have been removed.

---

### Decision · Program_Chairs · 2025-11-12

Accept